# Study of Chloride Ion Diffusion in Coral Aggregate Seawater Concrete with Different Water–Cement Ratios under Load

**DOI:** 10.3390/ma16020869

**Published:** 2023-01-16

**Authors:** Guangmin Dai, Qing Wu, Kailong Lu, Shiliang Ma, Wei Wang, Hao Zhou, Chenggong Cai, Zuocheng Han, Jiaming Chen

**Affiliations:** 1College of Civil Engineering and Architecture, Jiangsu University of Science and Technology, Zhenjiang 212114, China; 2Zhenjiang University Park Headquarters, Zhenjiang University, Zhenjiang 311122, China

**Keywords:** coral aggregate seawater concrete, load, water–cement ratio, chloride ion diffusion coefficient, COMSOL

## Abstract

This study was conducted to investigate the chloride ion transport in coral aggregate seawater concrete (CASC) with varying water–cement ratios under different loads. The ultimate compressive strength was obtained by conducting compression testing of three groups of CASC with different water–cement ratios. Steady loads of 0%, 10%, and 20% of their respective ultimate compressive strengths were applied to the concrete specimens with different water–cement ratios. After being subjected to a seawater erosion test for 30, 60, 90, 120, and 180 days, the chloride ion concentration at different depths was measured to determine the chloride ion diffusion coefficient. Meanwhile, the chloride ion diffusion coefficients of CASC were verified by comparing them with results obtained from numerical simulations performed using COMSOL software. The test results show that the internal pore space of CASC expands, leading to acceleration of the chloride ion transport rate when applied loads are increased. The initial chloride ion concentration of CASC rises as the water–cement ratio rises, and the concentration gradient formed with artificial seawater lowers, decreasing the chloride ion transport rate. When the water cement ratio decreases and the load increases, the diffusion coefficient increases. Using the numerical simulation method of COMSOL software, it was proved that the model has good applicability and accuracy in predicting chloride ion transport in CASC.

## 1. Introduction

Due to the long-term immersion of reinforced concrete structures in the marine environment, the chloride ions in seawater penetrate the protective layer of concrete and reach the surface of reinforcement, causing varying degrees of corrosion. Seawater corrosion reduces the durability and safety of structures. Chloride ion erosion has a significant influence on the service life of any structure, and it is one of the main causes of concrete structures’ degradation [1,2,3,4,5,6,7,8,9]. The average service life of reinforced concrete structures can drop to 25 years because of severe corrosion [10]. In the construction of structures on some islands, ordinary concrete materials are scarce. Transportation costs are too high. Construction times are too long. Local materials are needed. When raw materials are changed from ordinary concrete to CASC, the corrosion of CASC due to chloride ion attack becomes more severe due to the high chloride ion content in CASC’s raw material. When CASC is used for the aggregates, the concrete strength is reduced. This makes the ion transport under load more obvious. Thus, the durability of CASC becomes a problem that must be addressed [11,12,13,14,15,16,17,18,19,20].

When concrete structures are exposed to prolonged pressure in the marine environment, the concrete starts deteriorating. Sustained compressive loading and seawater erosion work together, wherein the damage accumulation generated by the loading action interferes with the transfer of harmful ions in concrete. A sharp drop in the durability performance of concrete is observed, affecting the normal use and service life of the structure. Wu et al. [21] found that the chloride ion concentration distribution decreases as compressive stress rises, whereas it increases with increasing exposure time. They also proposed a chloride ion transport model based on the stress ratio in concrete under holding pressure to assess the service life of concrete. Xu [22] derived the variation of chloride ion content in concrete at different depths based on Fick’s second law and obtained the diffusion coefficient of chloride ion. The results showed that load has a great influence on chloride ion transportability in concrete. Fu et al. [23] studied the chloride ion migration behavior of concrete under the coupling effect of water flow and load. The results showed that the thickness of the damage layer and chloride ion concentration increased with the age of erosion. Wang et al. [24] used a self-designed tidal circulation device and salt spray chamber to study the chloride ion concentration in the tidal and salt spray zones. They developed a prediction model for chloride ion concentration in concrete considering the effect of load level.

There remain certain discrepancies between experimental and real environments. Considerably, numerical simulation has been used to verify the experimental accuracy with real time data. Chen et al. [25] used finite element simulation to investigate the effects of the thickness of the interfacial transition zone, mortar and permeability properties of the interfacial transition zone on the distribution of the chloride ion concentration inside concrete. In comparison to other simulation methods, COMSOL can readily perform a variety of more complex multi-physical field coupling operations, as well as create mathematical expressions of partial differential equations and boundary conditions on the interface. Using simulations, Liu et al. [26] concluded that the concrete porosity changes due to load action, and the change in porosity leads to change in the chloride ion diffusion coefficient. They proposed a model to determine the chloride ion diffusion coefficient in concrete considering the effect of load action and verified the combination of COMSOL multi-physical field coupling algorithm with numerical modeling to simulate chloride ion diffusion in loaded concrete. Xu et al. [27] used a combination of simulation and experiments. They used COMSOL to develop fine-scale models of aggregates, mortar, interfacial transition zone, and crack and damage zone to study chloride ion migration. They validated the numerical model against experimental data and consequently explored the effect of crack self-healing on the chloride ion diffusion coefficient in concrete.

Although many researchers have studied chloride ion transport in ordinary concrete, limited studies have been performed considering CASC using COMSOL simulation, especially under the combined influence of loading conditions and water–cement ratios. In this study, the free chloride ion concentration in CASC under artificial seawater erosion and load closer to the actual working environment was investigated. The chloride ion diffusion coefficient proposed by Fick’s second law was used to explore the chloride ion diffusion law. On the other hand, COMSOL software was used to develop the chloride ion transport model, which was compared with the experimental data to validate its reliability. Finally, the results were used to provide data support for the next step of durability improvement of CASC.

## 2. Test Materials and Methods

### 2.1. Test Materials

The cement used in the study was 42.5 ordinary silicate cement, and the composition is listed in Table 1. Figure 1 shows the fine aggregate, which was coral sand that was well-graded and belonged to the second zone sand. Table 2 lists physical properties of the fine aggregate. Figure 2 shows the coarse aggregate, which was coral stone with particle sizes ranging from 5 to 20 mm in a continuous gradation. Table 3 lists the physical properties of the coarse aggregate. Water content was reduced by 25% by weight using a high-efficiency water reduction agent. As stated in Table 4, the mixing water was artificial seawater prepared according to ASTM D1141-2013 [28]. As stated in Table 5, the CASC’s design strength grade for this test was C30, with water–cement ratios of 0.30, 0.35, and 0.40.

### 2.2. Preparation of Specimens

Specimen sizes of 300 × 100 × 100 mm and 100 × 100 × 100 mm were used. The 300 × 100 × 100 mm specimens were tested for 30, 60, 90, 120, 150 and 180 days under artificial seawater erosion. All of the specimens were cast at the same time and demolded 24 h after casting, followed by standardized curing for 28 days. Concrete compressive strength tests were carried out according to the standard test methods of Physical and Mechanical Properties of Concrete (GBT50081-2019) [29]. Specimens of 100 × 100 × 100 mm dimensions were used to determine the ultimate compressive strength of CASC with different water–cement ratios in order to obtain more accurate loads for subsequent tests. The four 300 × 100 mm faces of the 300 × 100 × 100 mm specimens were coated with wax before loading, and the two opposing 100 × 100 mm faces were used as erosion surfaces.

### 2.3. Test Method

#### 2.3.1. Concrete Cube Compressive Strength Test

A YAW-2000B press device was used to determine the compressive strength of C30 concrete at 28 days. The compressive strength of a concrete cube can be calculated using the following equation:(1)$$f_{ce}=\frac{F}{A}$$
where *f_ce_* is the compressive strength of the cubic concrete specimen. *F* is the specimen damage load, and *A* is the pressure-bearing area of the specimen.

The concrete cube compressive strength was calculated to a precision of 0.1 MPa, and the arithmetic mean of the test results of the three specimens was taken as the strength value of the group of specimens.

#### 2.3.2. Test Method for Chloride Ion Concentration in CASC under Load

Self-designed equipment was used to load 100 × 100 × 300 mm cubic specimens immersed in artificial seawater, with the continuous load set at 0%, 10% and 20% of the compressive strength in Figure 3 and Figure 4. The test blocks were unloaded after 1, 2, 3, 4 and 6 months of compression. A hole was drilled to obtain chloride ion powder at different depths of the specimen, and the chloride ion concentration was determined using a rapid chloride ion test.

##### Sampling Method

As shown in Figure 5, the drilling method was chosen to drill the powder at the midline of both sides of the specimen. A hand drill of 6 mm diameter bit was used herein. The sampling depths were 15, 30, 45, 60, and 75 mm. The collected powder mass of each layer of the specimen was ensured to be 5 g, and the extra powder in the hole was meticulously cleaned using a brush. The powder was sieved to pass through a 0.15 mm size sieve to remove the larger particles. The powder was dried in an oven for 2 h and cooled before being placed in a triangle flask with 20 mL of distilled water and thoroughly shaken. The triangular flask was covered with a flat dish and placed on a heating tool to boil for 5 min. The heat source was then turned off, the flask was closed, and the triangle flask was placed for 24 h. The solution was extracted after being filtered via a fast quantitative filter paper.

##### Test Method

After obtaining the filtrate, a rapid chloride ion content device was used to detect the chloride ion concentration (*C*) within the filtrate, and the ratio of chloride ion to the mass of cementitious material within the hardened concrete was calculated using the equation,
(2)$$C_{f}\left(\%\right)=\frac{35.5\times C\times V}{G}\times \frac{m_{B}+m_{S}+m_{W}}{m_{B}}\times 100\%$$
where *C_f_* is the ratio of chloride ion to mortar mass in concrete (%); *C* is the concentration of chloride ion in the filtrate in terms of substance (mol/L); *V* is the volume of the solution for extracting chloride ion (mL); *G* is the mortar sample quality (g); *m_B_* is the amount of cement used in 1 m^3^ of concrete in the concrete mix ratio (kg); *m_S_* is the amount of sand used in 1 m^3^ of concrete in the concrete mix ratio (kg); *m_w_* is the amount of water used in 1 m^3^ of concrete in the concrete mix ratio (kg).

This experiment was conducted to determine the free chloride ion content (*C_f_*) of concrete powder at different depths in accordance with the national industry standard mentioned in the text of technical specifications for chloride ion content in concrete (JGJ/T 322-2013) [30]. Figure 6 shows the model SSWY-810 chlorine ion rapid analyzer used in the experiment.

##### Microscopic Testing

The sample was first prepared, then sprayed with gold. The observation surface of the sample was connected to the base of an electron microscope by conductive adhesive tape.

#### 2.3.3. Calculation Method for Parameters

(1)Surface free chloride ion content and chloride ion diffusion coefficient

Regression analysis was used to fit the quadratic relationship between the depths and the average values of *C_f_* in the test to obtain the average values of *C_f_* at depths of 15 mm, 30 mm, 45 mm, 60 mm, and 75 mm. Using this relationship, the value of *C_s_* was calculated when the depth *x* was 0.

Since the concrete specimens were intact in the test, and the erosion stage was at constant temperature; the only factor affecting the chloride ion diffusion coefficient was time (t). Hence, the following Equation (3) was used to fit the relationship curve between the free chloride ion concentration in the test study and the depth *x* in the Origin program to obtain the chloride ion diffusion coefficient,
(3)$$C_{f}=C_{0}+\left(C_{s}-C_{0}\right)\left(1-erf\frac{x}{2\sqrt{Dt}}\right)$$
where *C*_0_ is the raw chloride ion content in concrete (%); *C_f_* is the free chloride ion content of concrete (%); *C_s_* is the concrete surface free chloride ion content(%); *D* is the chloride ion diffusion coefficient of concrete (m^2^·s^−1^); and *erf* is the error function.

(2)Time dependence of chloride ion diffusion coefficient

Since the hydration phase of concrete is slow and the void structure in concrete improves owing to continuous hydration, which lowers the *D* value, it was found that *D* is dependent on time. After obtaining the chloride ion diffusion coefficient for CASC, the reduction of *D* with time was studied and expressed as the power function below [31],
(4)$$D\left(t\right)=A\cdot t^{-m}$$
where *D (t)* is the diffusion coefficient of chloride ion in exposure time *t* (m^2^·s^−1^). *t* is the exposure time (s). *m* is the attenuation coefficient of the chloride ion diffusion coefficient.

(3)Time dependence of surface free chloride ion content

The free chloride ion content on the concrete surface gradually grew as the erosion time increased. After obtaining the chloride ion content at different times, the data could be fitted and expressed as the following power function,
(5)Cs=a−be−tc
where *C_s_* is the apparent chloride ion content at exposure time *t* (%); *t* is the exposure time *t* (s), and *a*, *b*, and *c* are the fitting parameters.

## 3. Results and Discussions

### 3.1. Ultimate Compressive Strength of CASC

The compressive strength results for C30 CASC at different water–cement ratios are shown in Table 6.

The strength value of each group was calculated using the arithmetic mean of the measured results of three specimens with varying water–cement ratios.

### 3.2. Distribution Pattern of Chloride Ion Content

The initial chloride ion concentration (*C*_0_) and free chloride ion concentration (*C_f_*) were obtained from the experiment. The apparent chloride ion concentration (*C_s_*) was obtained by *C_f_* fitting, and finally, the diffusion coefficient (*D*) was obtained from the fitting of *C*_0_, *C_f_* and *C_s_*.

#### 3.2.1. The Free Chloride Ion Concentration (*C_f_*)

##### The Chloride Ion Transport Law with the Same Water–Cement Ratio and Load

The *C_f_* of CASC was determined for 30, 60, 90, 120, 150 and 180 days of the test with different test environments. The values of *C*_0_ were about 0.3216%, 0.4204%, and 0.5574% for the water–cement ratios of 0.30, 0.35, and 0.40, respectively.

As shown in Figure 7, when the water–cement ratio was 0.30 and the load was 0, the *C_f_* increased by 9.05~65.08% after 30 days of seawater exposure compared with *C_0_*, 0.30. After seawater erosion from 60 days to 120 days, the *C_f_* increased by 6.89~25.88% compared to that recorded 30 days earlier. After 180 days of seawater erosion, *C_f_* increased by 11.02~17.81% compared with the *C_f_* at 120 days. When the water–cement ratio was 0.30 and the load was 0.1, the *C_f_* increased by 18.81~124.38% compared with *C_0_*, 0.30 after 30 days of seawater erosion. After seawater erosion from 60 days to 120 days, the *C_f_* increased by 4.80~23.09% compared to 30 days earlier. After 180 days of seawater erosion, the *C_f_* increased by 14.20~22.05% compared with the *C_f_* at 120 days. When the water–cement ratio was 0.30 and the load was 0.2, the *C_f_* increased by 27.43~132.49% after 30 days of seawater erosion compared with *C_0_*, 0.30. From 60 days to 120 days of seawater erosion, the *C_f_* increased by 10.51~34.85% compared with 30 days earlier. After 180 days of seawater erosion, the *C_f_* increased by 6.20~30.56% compared with the *C_f_* at 120 days.

After 30 days of seawater erosion, the largest rise in *C_f_*, from 65.08% to 132.49%, was noticed when the water–cement ratio was 0.30 and the load was changed from 0 to 0.2. From 60 days to 120 days of seawater erosion, the maximum increase in *C_f_* was from 23.09% to 34.85%. The maximum increase in *C_f_* was from 17.81% to 30.56% after 180 days of seawater erosion.

As shown in Figure 8, when the water–cement ratio was 0.35 and the load was 0, the *C_f_* increased by 2.62~35.68% after 30 days of seawater erosion compared with the initial chloride ion concentration at the water–cement ratio of 0.35 (*C*_0_, 0.35). After seawater erosion from 60 days to 120 days, the *C_f_* increased by 8.04~23.86% compared to the *C_f_* of 30 days earlier. After 180 days of seawater erosion, *C_f_* increased by 17.85~29.18% compared to the *C_f_* at 120 days. When the water–cement ratio was 0.35 and the load was 0.1, *C_f_* increased by 9.68~70.43% after 30 days of seawater erosion compared with *C*_0_, 0.35. After seawater erosion from 60 days to 120 days, *C_f_* increased by 5.72~31.03% compared with 30 days earlier. After 180 days of seawater erosion, *C_f_* increased by 17.29~25.30% compared with the *C_f_* at 120 days. When the water–cement ratio was 0.35 and the load was 0.2, the *C_f_* increased by 20.65~108.25% compared with *C*_0_, 0.35 after 30 days of seawater erosion. After seawater erosion from 60 days to 120 days, *C_f_* increased by 10.51~22.53% compared to the *C_f_* of 30 days earlier. After 180 days of seawater erosion, the *C_f_* increased by 10.51~22.53% compared with the *C_f_* at 120 days.

When the water–cement ratio was 0.35 and the load was changed from 0 to 0.2, the maximum increment of *C_f_* increased from 35.68% to 108.25% after 30 days of seawater erosion. From 60 days to 120 days of seawater erosion, the maximum increase in *C_f_* was from 22.92~31.03%. The maximum increase in *C_f_* was from 22.53~29.18% after 180 days of seawater erosion.

As shown in Figure 9, when the water–cement ratio was 0.40 and the load was 0, the *C_f_* increased by 1.34~25.27% after 30 days of seawater erosion compared with the initial chloride ion concentration of 0.4204%. After seawater erosion from 60 days to 120 days, the *C_f_* increased by 2.55~19.39% compared to the *C_f_* of 30 days earlier. After 180 days of seawater erosion, *C_f_* increased by 7.67~17.06% compared to the *C_f_* at 120 days. When the water–cement ratio was 0.40 and the load was 0.1, *C_f_* increased by 6.72~56.35% after 30 days of seawater erosion compared with *C*_0_, 0.40. After seawater erosion from 60 days to 120 days, *C_f_* increased by 7.35~34.03% compared with 30 days earlier. After 180 days of seawater erosion, *C_f_* increased by 4.94~16.11% compared with the *C_f_* at 120 days. When the water–cement ratio was 0.40 and the load was 0.2, the *C_f_* increased by 13.73~77.86% compared with *C*_0_, 0.40 after 30 days of seawater erosion. After seawater erosion from 60 days to 120 days, *C_f_* increased by 6.13~36.62% compared to the *C_f_* of 30 days earlier. After 180 days of seawater erosion, the *C_f_* increased by 14.96~27.93% compared with the *C_f_* at 120 days.

When the water–cement ratio was 0.40 and the load changed from 0 to 0.2, the maximum increment of *C_f_* was from 25.27% to 77.86% after 30 days of seawater erosion. From 60 days to 120 days of seawater erosion, the maximum increase in *C_f_* was from 13.34~36.62%. The maximum increase in *C_f_* was from 16.11~27.93% after 180 days of seawater erosion.

Figure 7, Figure 8 and Figure 9 indicate that the free chloride ion concentration (*C_f_*) of CASC under the same water–cement ratio and load is positively correlated with the test time. The *C_f_* increased with the increase in test time and load; however, it decreased with the increase in diffusion depth. The growth interval showed that the growth of *C_f_* was fast after 30 days of seawater erosion, and then the growth was flat.

##### The Chloride Ion Transport Law under the Same Water–Cement Ratio and Different Loads

As shown in Figure 10a with the initial chloride ion concentration (*C*_0,0.30_) for a water–cement ratio of 0.30, it can be concluded that when the water–cement ratio was 0.30 and the load was 0 at the 180 day mark, the *C_f_* at each depth improved by 71.42~177.55% compared to *C*_0,0.30_. When the water–cement ratio was 0.30 and the load was 0.1 after 180 days of seawater erosion, *C_f_* improved by 123.54~276.93% compared to *C*_0,0.30_. When the water–cement ratio was 0.30 and the load was 0.2, *C_f_* increased by 165.58~392.51% compared to *C*_0,0.30_ after 180 days of seawater erosion.

As shown in Figure 10b with the initial chloride ion concentration (*C*_0,0.35_) for a water–cement ratio of 0.35 and load at 0, it was concluded that *C_f_* at each depth improved by 59.25~150.15% after 180 days of seawater erosion as compared with *C*_0,0.35_. When the water–cement ratio was 0.30 and the load was 0.1 at the 180 day mark in seawater, *C_f_* increased by 77.28~237.18% compared to *C*_0,0.35_ after erosion. When the water–cement ratio was 0.30 and the load was 0.2, *C_f_* increased by 145.41~316.44% compared to *C*_0,0.35_ after 180 days of seawater erosion.

As shown in Figure 10c with the initial chloride ion concentration (*C*_0,0.40_) for a water–cement ratio of 0.40, it was concluded that for *C_f_* at each depth, when the water–cement ratio was 0.40 and the load was 0, after 180 days of seawater erosion, *C_f_* increased by 22.48~126.9% compared to *C*_0,0.40_. When the water–cement ratio was 0.40 and the load was 0.1, after 180 days of seawater erosion, *C_f_* increased by 83.85~196.81% compared to *C*_0,0.40_. When the water–cement ratio was 0.40 and the load was 0.2, after 180 days of seawater erosion, *C_f_* increased by 83.85~196.81% compared to *C*_0,0.40_. When the water–cement ratio was 0.40 and the load was 0.2, *C_f_* increased by 122.5~252.32% compared to *C*_0,0.40_ after 180 days of seawater erosion.

From Figure 10, it can be seen that the *C_f_* of CASC at the same water–cement ratio and different loads was proportional to the load. For the same erosion time, *C_f_* increased with the increase in load. For a water–cement ratio of 0.30, the rate of increase in *C_f_* at load 0.1 was 1.56 times that at load 0, and at a load of 0.2, the rate of increase in *C_f_* was 1.42 times that at the load of 0.1. For a water–cement ratio of 0.35, the rate of increase in *C_f_* at load 0.1 was 1.58 times that at load 0, and the rate of increase in *C_f_* at load 0.2 was 1.33 times that at load 0.1. The rate of increase in *C_f_* at load 0.1 was 1.55 times that at load 0, and the rate of increase in *C_f_* at load 0.2 was 1.28 times that at load 0.1 when the water–cement ratio was 0.40. It was concluded that when the load changed from 0 to 0.1, the *C_f_* increase rate was stable. When the load changed from 0.1 to 0.2, the water–cement ratio increased and the *C_f_* increase rate decreased.

In a full immersion environment of artificial seawater, the pores are fully saturated, and chloride ions are solely transported by means of diffusion. The chloride ion transport depends on the concentration gradient formed by chloride ions inside the specimen and chloride ions from seawater. A lower water–cement ratio means less mixed seawater, resulting in a CASC specimen with lower *C_0_* and a larger chloride ion concentration gradient with the erosion environment. Thus, the chloride ion intrusion at a low water–cement ratio will be faster. The load will increase the porosity of the CASC specimen, making chloride ion transport easier. The greater the load, the greater the porosity of the CASC specimen, and the faster the transfer rate.

As the diffusion continues, the CASC becomes more fully hydrated internally. The generation of hydration products such as calcium silicate hydrate (CSH gel) blocks transmission channels, while the invasion of various ions will cause the blockage of chloride ion diffusion channels, thus improving the resistance of CASC to chloride ion penetration and consequently stabilizing the subsequent erosion process.

#### 3.2.2. Calculation of Apparent Chloride Concentration (*C_S_*)

According to the calculation method used for the parameters in Equation (3), the fitted curve where the apparent chloride ion concentration (*C_S_*) is derived is shown in Table 7.

The goodness of fit of the fitted curve was 0.97481~0.99988, which is a good fit. The value of the function was the apparent chloride ion concentration (*C_S_*) when the time of the fitted curve was 0.

As shown in Table 8, the apparent chloride ion concentration (*C_S_*) of CASC with the same water–cement ratio and the same load was positively correlated with the test time. The values of *C_S_* increased with the increases in test time and load.

#### 3.2.3. Chloride Ion Diffusion Coefficient (*D*)

##### Calculation and Analysis of Chloride Ion Diffusion Coefficient (*D*)

Based on *C_f_* and *C_S_*, the chloride ion diffusion coefficient (*D*) was calculated using Fick’s second law in Equation (1), as shown in Table 9.

As shown in Figure 11a, when the water–cement ratio was 0.30 and the load was 0, the rate of change in *D* varied from 80.30% to 85.65% as the test time varied from 30 to 180 days. When the water–cement ratio was 0.30 and the load was 0.1, the rate of change in *D* varied from 79.39% to 87.04%. When the water–cement ratio was 0.30 and the load was 0.2, the rate of change in *D* varied from 75.75% to 88.35%. When the water–cement ratio was 0.35 and the load was 0, the rate of change in *D* varied from 83.41% to 93.17%. When the water–cement ratio was 0.35 and the load was 0.1, the rate of change in *D* varied from 79.97% to 87.50%. When the water–cement ratio was 0.35 and the load was 0.2, the rate of change in *D* varied from 82.50% to 81.61%. When water–cement ratio was 0.40 and the load was 0, the rate of change in *D* varied from 64.90% to 89.58%. The rate of change in *D* varied from 81.85% to 91.38% when the water–cement ratio was 0.40 and the load was 0.1, whereas it varied from 86.58% to 88.64% when the water–cement ratio was 0.40 and the load was 0.2.

Figure 11 shows that the diffusion coefficient *D* of CASC was inversely proportional to the test time for the same water–cement ratio and different loads and that *D* declined as the test duration increased. Under the same water–cement ratio, the *D* of CASC dropped faster in the early stages and slower as the test duration progressed. As the load increased, the diffusion coefficient *D* dropped more slowly. The diffusion coefficient *D* fell gradually as the water–cement ratio increased. The total amount of various hydration products formed inside the capillary pores increased as the concrete hydrated gradually, reducing the pore size and porosity. The hydration products formed quickly in the early stage, making the drop in diffusion coefficient *D* quicker in the early stage and slower in the later stage. At the same time, as diffusion continued, the gradient of concentration formed by chloride ions inside the specimen and chloride ions in the seawater gradually decreased, and diffusion slowed down.

Taking the coral concrete sample with water–cement ratio of 0.35 as an example, the microscopic morphological structure is shown in Figure 12. From the figure, it can be found that the hydration products increased sequentially, thus blocking the chloride ion transport channels, and D decreased gradually.

##### Chloride Ion Diffusion Coefficient (*D*) Fitting

The data from Table 9 were fitted and compared using the Origin program to derive an expression for the chloride diffusion coefficient (*D*) as a function of test time, with the fitted functions as shown in Table 10.

The goodness-of-fit of the fitted curves ranged from 0.99836 to 0.99859, and the fit was found to be good.

### 3.3. COMSOL Software Chloride Ion Transport Simulation

#### 3.3.1. Transport Model

(1)Theoretical equations of chloride ion transport

When the chloride ion diffusion model is one-dimensional diffusion, the theoretical equation for the transport of chloride ion in CASC along the *x*-direction is obtained by Fick’s second law of diffusion, as follows,
(6)$$C_{f}=C_{0}+\left(C_{s}-C_{0}\right)\left(1-erf\frac{x}{2\sqrt{Dt}}\right)$$

(2)Initial conditions

t = 0, the initial chloride ion concentrations in CASC (*C*_0_) are:(7)$$C_{0,0.30}=0.3216\%$$
(8)$$C_{0,0.35}=0.4204\%$$
(9)$$C_{0,0.40}=0.5579\%$$

(3)Boundary conditions and oxygen diffusion concentration

Assuming that chloride ion transport is one-dimensional, the CASC model was subjected to ion transport on one side, and the remaining three sides were flux-free. The free chloride ion concentration content in CASC was measured in the experimental phase as a function of time. By Equation (6), the apparent chloride ion concentration in CASC with time could be obtained from different loads as well as water–cement ratios, as shown in the equations from Equation (10) to Equation (18):(10)Cs,0.30,0=0.01118−0.0065e−t10061300
(11)Cs,0.30,0.1=0.02064−0.01351e−t23015700
(12)Cs,0.30,0.2=0.02086−0.01603e−t9275530
(13)Cs,0.35,0=0.02421−0.01952e−t28566900
(14)Cs,0.35,0.1=0.02082−0.01621e−t9915150
(15)Cs,0.35,0.2=0.03044−0.02369e−t19083800
(16)Cs,0.40,0=0.01867−0.0141e−t11025700
(17)Cs,0.40,0.1=0.02159−0.017e−t6853770
(18)Cs,0.40,0.2=0.02472−0.01719e−t9255250

(4)Chloride ion diffusion coefficient

The chloride ion diffusion coefficient *D* during chloride ion transport in CASC with different water–cement ratios under load can be found by employing the equations from Equation (19) to (27),
(19)$$D_{0.30,0}\left(t\right)=9.368\times 10^{-8}\cdot t^{-0.36324}$$
(20)$$D_{0.30,0.1}\left(t\right)=1.24\times 10^{-7}\cdot t^{-0.39218}$$
(21)$$D_{0.30,0.2}\left(t\right)=6.68\times 10^{-8}\cdot t^{-0.35438}$$
(22)$$D_{0.35,0}\left(t\right)=4.00\times 10^{-8}\cdot t^{-0.33245}$$
(23)$$D_{0.35,0.1}\left(t\right)=1.24\times 10^{-7}\cdot t^{-0.39218}$$
(24)$$D_{0.35,0.2}\left(t\right)=1.50\times 10^{-7}\cdot t^{-0.3845}$$
(25)$$D_{0.40,0}\left(t\right)=6.03\times 10^{-6}\cdot t^{-0.68085}$$
(26)$$D_{0.40,0.1}\left(t\right)=6.68\times 10^{-8}\cdot t^{-0.35438}$$
(27)$$D_{0.40,0.2}\left(t\right)=2.28\times 10^{-8}\cdot t^{-0.26446}$$

Table 11 summarizes the parameters required to establish COMSOL numerical simulations with various water–cement ratios and free chloride ion concentrations under various loads based on the above conditions.

#### 3.3.2. Analysis of COMSOL Results

The established CASC model was simulated numerically using COMSOL software, combined with a series of conditions such as initial conditions, boundary conditions and chloride ion diffusion coefficients. A comparison of the experimental results and model values for chloride ion concentrations in CASC under loads 0, 0.1, and 0.2 and water–cement ratios 0.30, 0.35, and 0.40 was performed and is presented in Figure 13, Figure 14 and Figure 15.

It can be observed from the model results in Figure 13, Figure 14 and Figure 15 that the *C_f_* of CASC was positively correlated with the test time, and *C_f_* increased as the test time increased. With increasing test time and decreased diffusion depth, *C_f_* values increased. They were proportional to the load, and as the load grew, so did C_f_. At the same time, the curve pattern obtained from the simulation was in general agreement with the trend of the experimentally measured curve pattern; thus, it can be concluded that the model has good applicability.

The simulation of chloride transport in CASC could be easily and efficiently implemented by means of COMSOL software, and the calculated values of the model and the experimental data had a high correlation.

## 4. Conclusions

In this paper, the compressive strength of CASC with different water–cement ratios was determined, and the chloride ion transport law for CASC with different water–cement ratios under different loads was comparatively studied. The chloride ion diffusion in CASC was also simulated using COMSOL software and compared with physical test results. The following conclusions were obtained.

(1)The applied load has a significant impact on the performance of CASC. With the increase in applied load, CASC internal pores expanded, leading to the acceleration of chloride ion transport.(2)The water–cement ratio not only affects the strength of CASC, but also has a direct impact on chloride ion transport. Variation in the water–cement ratio altered the initial chloride ion concentration in CASC, producing different concentration gradients. A decrease in the water–cement ratio decreased the initial chloride ion concentration; however, it increased the concentration gradient and the transport speed.(3)Under the combined study of the water–cement ratio and load, when the water–cement ratio was decreased, the strength of CASC improved, while the diffusion coefficient decreased at a faster rate under the same load. Conversely, when the water–cement ratio was increased, the strength of CASC decreased, while the diffusion coefficient decreased at a slower rate under the same load.(4)Numerical simulation methods in COMSOL software were applied to the established chloride ion transport model for CASC, and it was concluded that the calculated model values correlated well with the physical test values. The variation curve for the chloride ion concentration with time was in good agreement with the simulated data curve. It was proved that the model has good applicability and accuracy for simulating chloride ion transport in CASC and simultaneously clarifying the chloride ion transport process in CASC.

## Figures and Tables

**Figure 1 materials-16-00869-f001:**
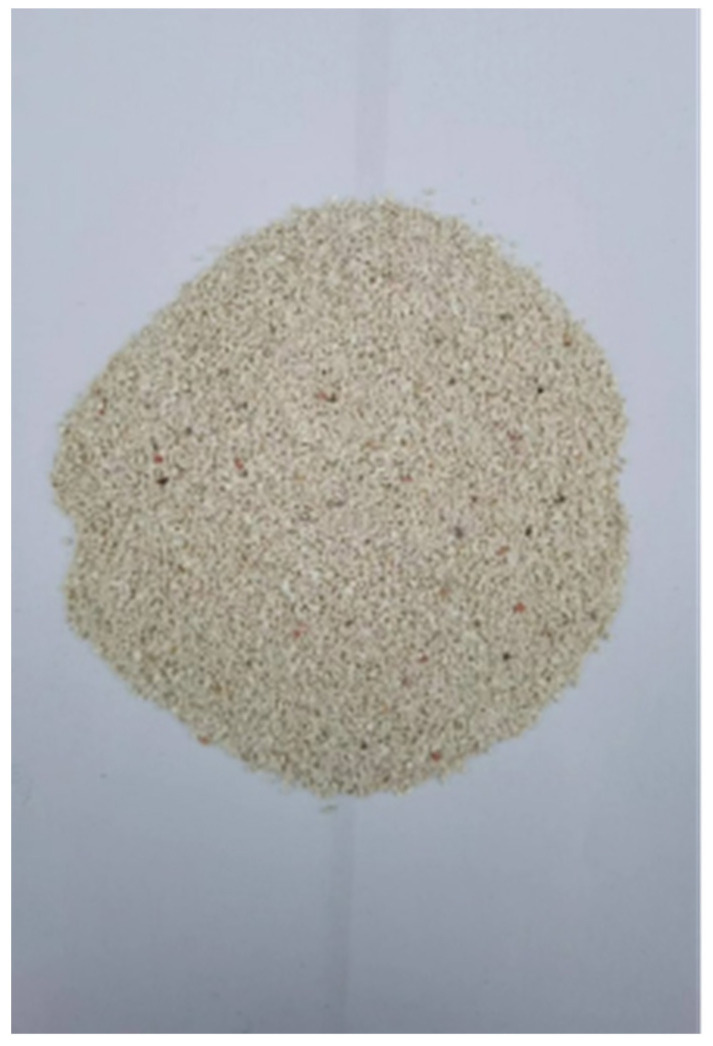
Fine aggregate.

**Figure 2 materials-16-00869-f002:**
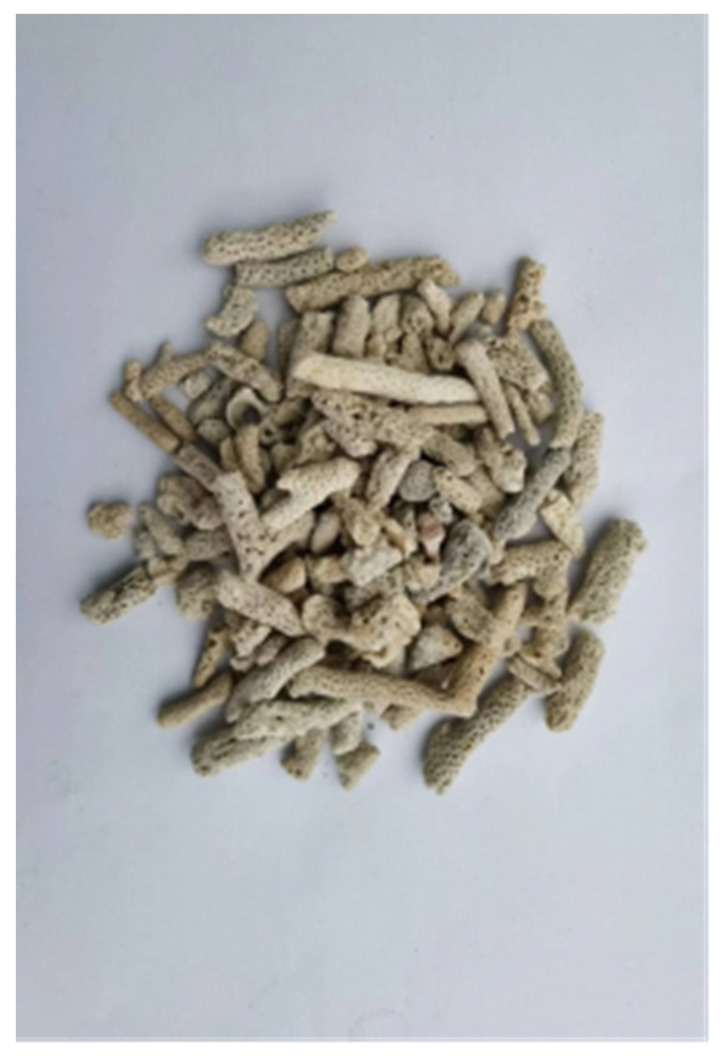
Coarse aggregate.

**Figure 3 materials-16-00869-f003:**
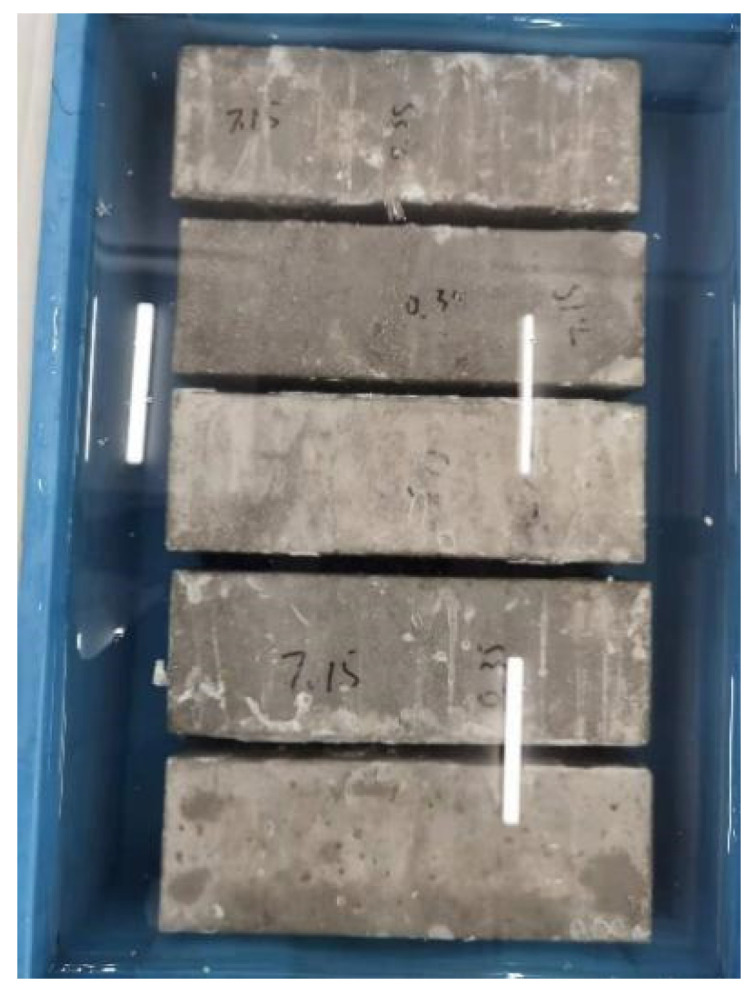
No load, seawater erosion specimen.

**Figure 4 materials-16-00869-f004:**
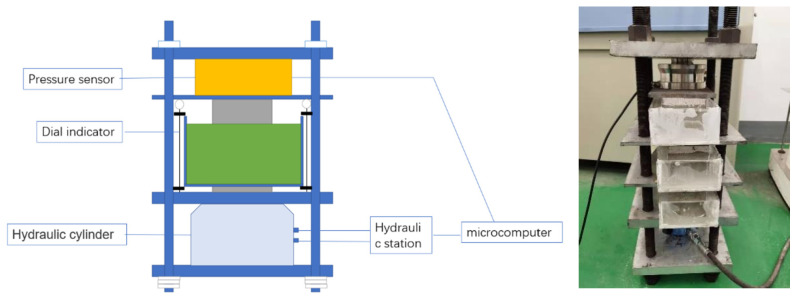
Continuous pressurized load and seawater joint action test equipment.

**Figure 5 materials-16-00869-f005:**
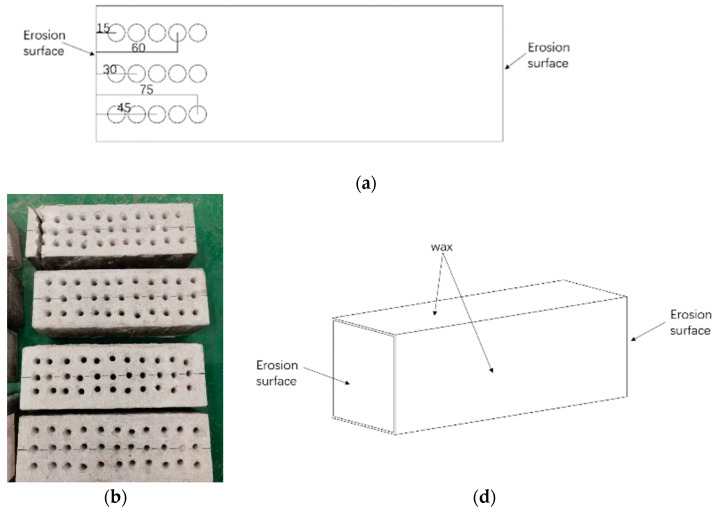
Drill hole for powder specimen (**a**–**c**).

**Figure 6 materials-16-00869-f006:**
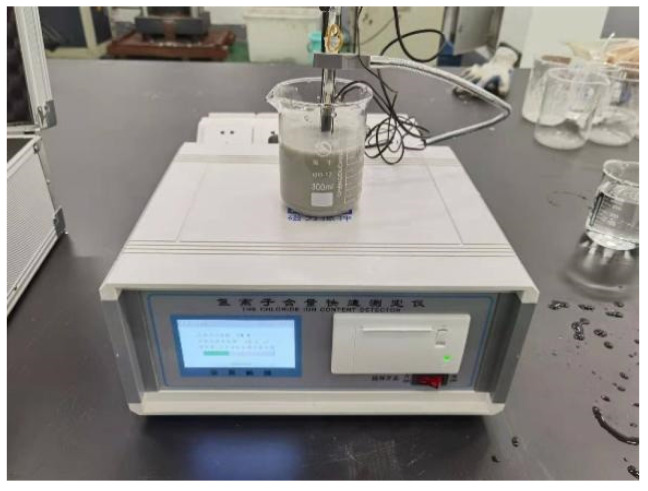
Chlorine ion fast tester.

**Figure 7 materials-16-00869-f007:**
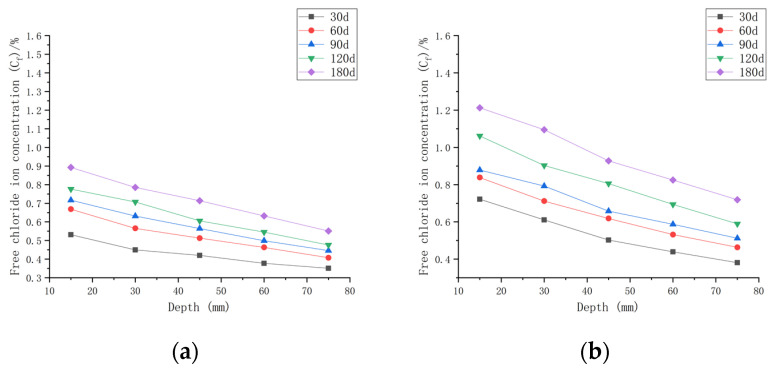
Relationship between free chloride ion concentration and depth of CASC with water–cement ratio 0.30 at different test times and loads. (**a**) Load is 0; (**b**) load is 0.1; (**b**) load is 0.1.

**Figure 8 materials-16-00869-f008:**
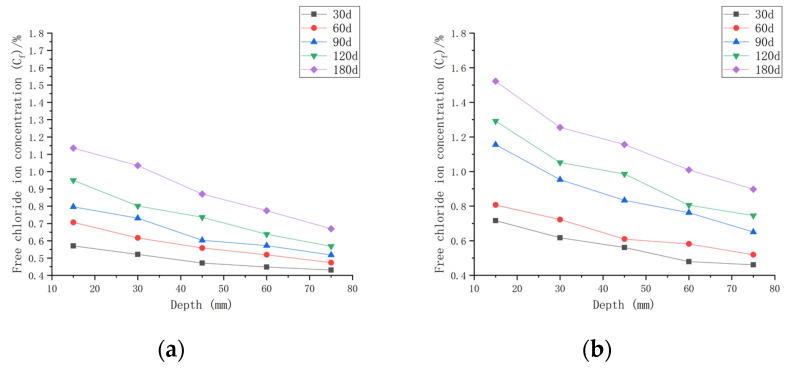
Relationship between free chloride ion concentration and depth of CASC with water–cement ratio of 0.35 at different test times and loads. (**a**) Load is 0; (**b**) load is 0.1; (**c**) load is 0.2.

**Figure 9 materials-16-00869-f009:**
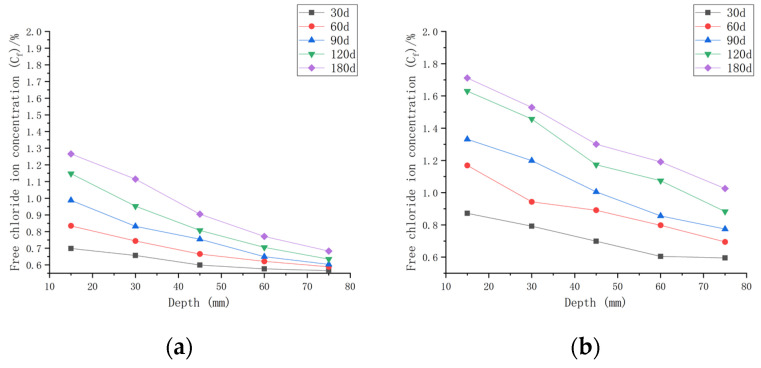
Relationship between free chloride ion concentration and depth of CASC with water–cement ratio 0.40 at different test times and loads. (**a**) Load is 0; (**b**) load is 0.1; (**c**) load is 0.2.

**Figure 10 materials-16-00869-f010:**
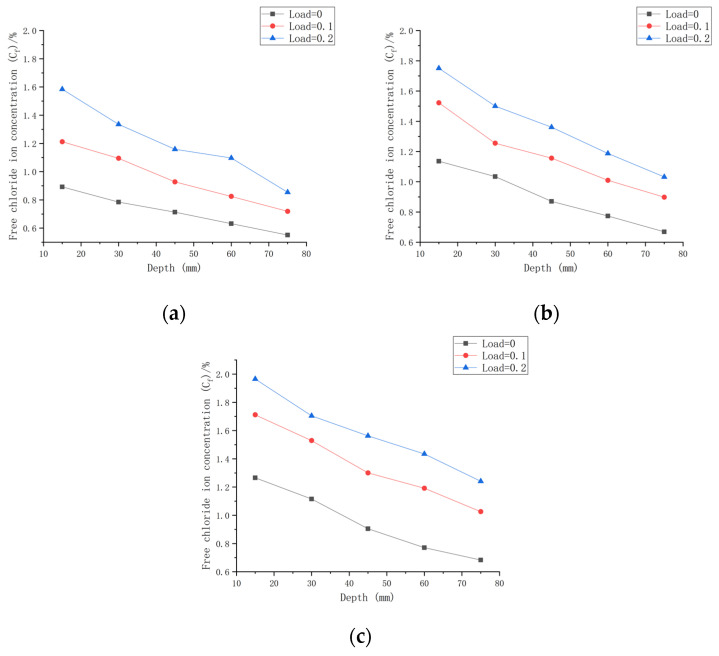
Relationship between chloride ion content and depth for different loads on CASC with the same water–cement ratio at 180 d. (**a**) W/C = 0.30; (**b**) W/C = 0.35; (**c**) W/C = 0.40.

**Figure 11 materials-16-00869-f011:**
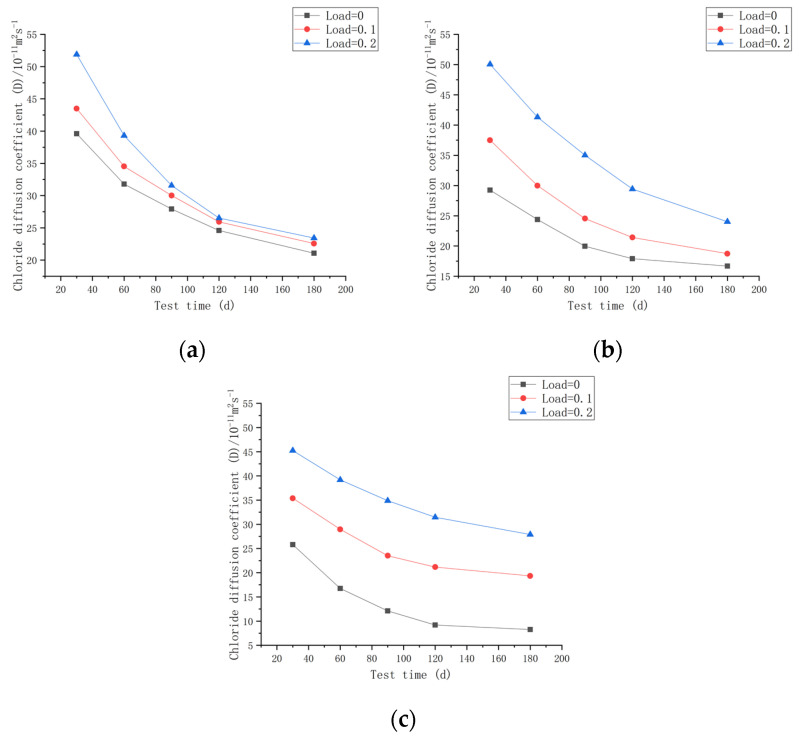
Relationship between D and time for different loads on CASC with the same water–cement ratio at 180 d. (**a**) W/C = 0.30; (**b**) W/C = 0.35; (**c**) W/C = 0.40.

**Figure 12 materials-16-00869-f012:**
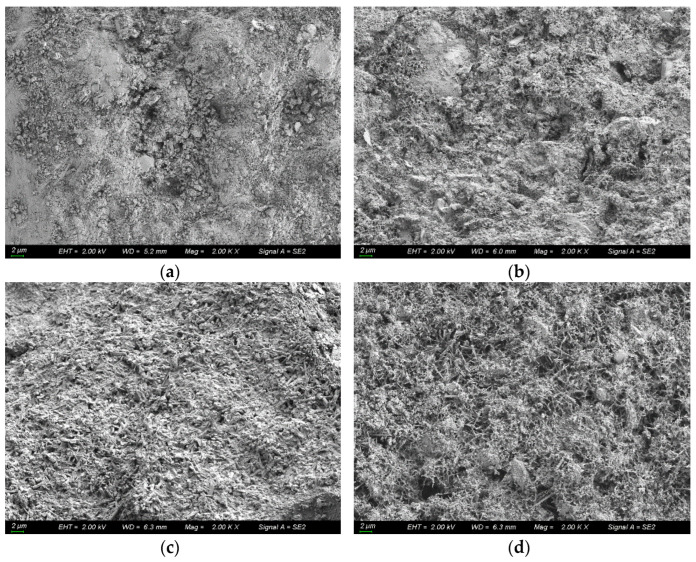
Microscopic morphological structure of coral concrete with water–cement ratio of 0.35. (**a**) t = 30 d; (**b**) t = 60 d; (**c**) t = 90 d; (**d**) t = 180 d.

**Figure 13 materials-16-00869-f013:**
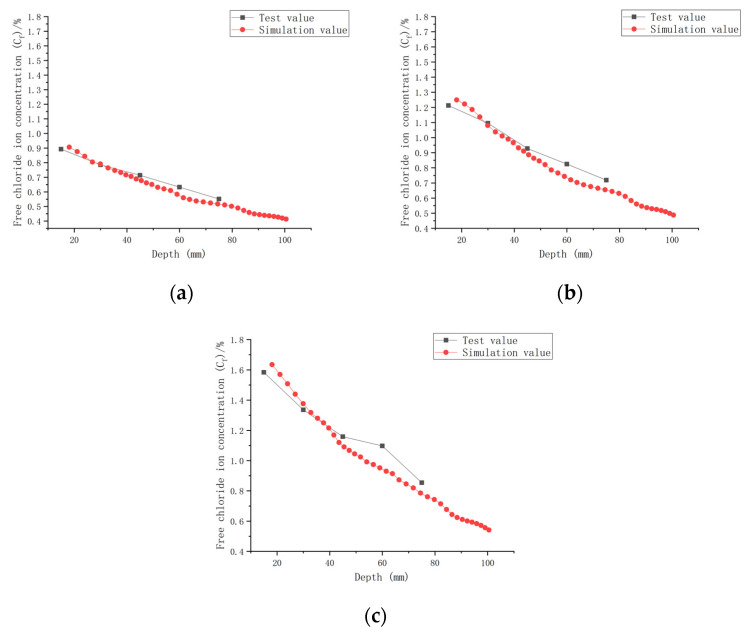
Relationship between *C_f_* and depth (W/C = 0.30). (**a**) Load is 0; (**b**) load is 0.1; (**c**) load is 0.2.

**Figure 14 materials-16-00869-f014:**
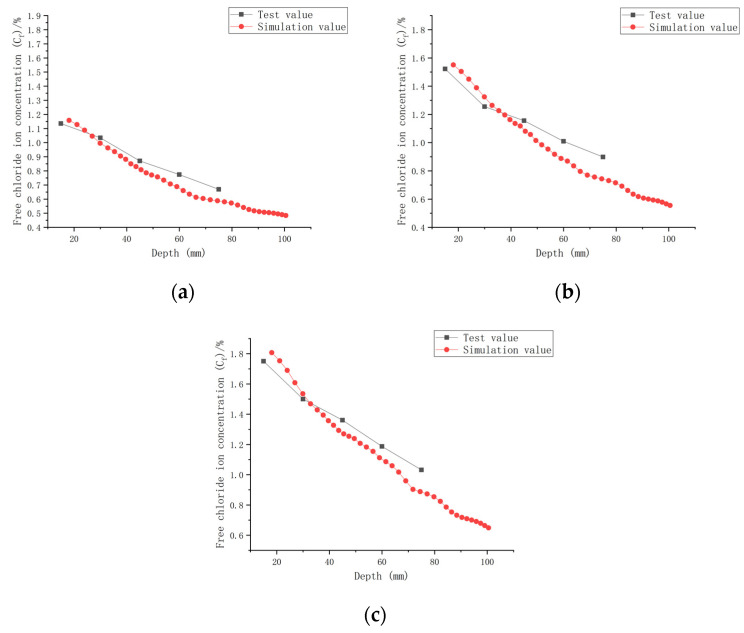
Relationship between *C_f_* and depth (W/C = 0.35). (**a**) Load is 0; (**b**) load is 0.1; (**c**) load is 0.2.

**Figure 15 materials-16-00869-f015:**
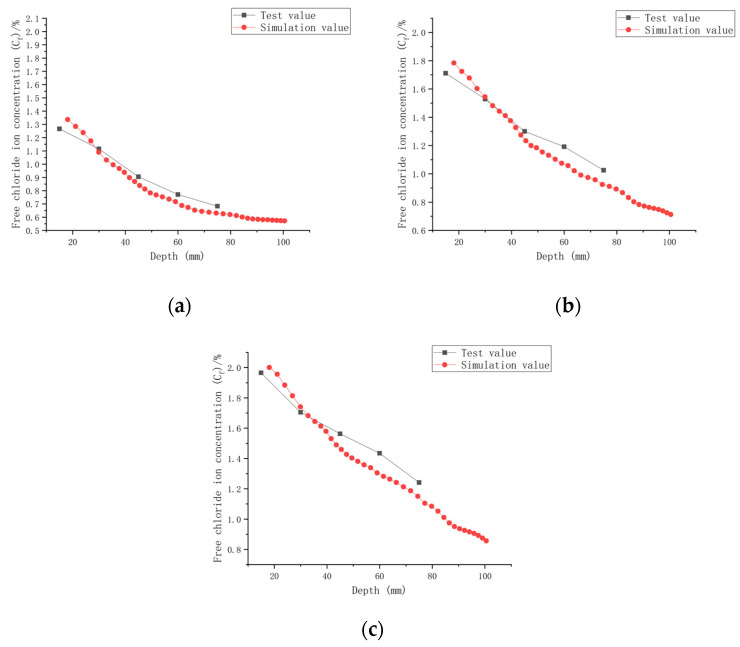
Relationship between *C_f_* and depth (W/C = 0.40). (**a**) Load is 0; (**b**) load is 0.1; (**c**) load is 0.2.

**Table 1 materials-16-00869-t001:** Chemical composition of ordinary silicate cement.

Components	SiO_2_	Al_2_O_3_	Fe_2_O_3_	CaO	M_g_O	SO_3_	Cl^−^	LOSS
Content (%)	24.99	8.26	4.03	51.42	3.71	2.51	0.043	3.31

**Table 2 materials-16-00869-t002:** Physical properties of fine aggregate.

Apparent Density kg/m^3^	Stacking Density kg/m^3^	Cylinder Compression Strength MPa	Porosity %	Natural Water Content %	1 h Water Absorption Rate%	Mud Content %
1865	928	1.6	55	0.1	17.1	0.58

**Table 3 materials-16-00869-t003:** Physical properties of coarse aggregate.

Apparent Density kg/m^3^	Stacking Density kg/m^3^	Void Ratio %	Natural Water Content %	1 h Water Absorption Rate %	Mud Content %
2450	1163	48	0.3	13.2	0.58

**Table 4 materials-16-00869-t004:** Chemical composition of artificial seawater.

Name	Molecular Formula	Quality (g/L)
Sodium chloride	NaCl	46.934
Magnesium chloride	MgCl_2_	9.962
Sodium sulfate	Na_2_SO_4_	7.834
Calcium chloride	CaCl_2_	2.204
Potassium chloride	KCl	1.328

**Table 5 materials-16-00869-t005:** Coral aggregate seawater concrete mix design.

Concrete Strength Grade	Water– Cement Ratio	Artificial Seawater (kg/m^3^)	Cementitious Material (kg/m)	Coral Sand (kg/m^3^)	Coral Reef (kg/m^3^)	Water Reducing Agent (kg/m^3^)
C30	0.30	167	557	765	762	4
C30	0.35	195	557	749	749	4
C30	0.40	223	557	733	736	4

Note: Due to the porous nature of coral, the coral aggregate needs to be pre-soaked until saturation before it is used to make CASC. This introduces a pre-absorption parameter for water = (coral reef + coral sand) × 8%.

**Table 6 materials-16-00869-t006:** Compressive strength of CASC.

	W/C = 0.30			W/C = 0.35			W/C = 0.40		
Compressive strength/MPa	38.0	33.2	35.8	37.1	34.2	31.2	34.2	30.3	36.1
Average strength/MPa	35.67			34.17			33.53		

**Table 7 materials-16-00869-t007:** Results and goodness of curve fitting (*C_S_*).

Number	a	b	c	Relevance
*C_s_* _,0.30,0_	0.01118	−0.0065	10,061,300	0.99297
*C_s_,* _0.30,0.1_	0.02064	−0.01351	23,015,700	0.99971
*C_s_,* _0.30,0.2_	0.02086	−0.01603	9,275,530	0.99057
*C_s_,* _0.35,0_	0.02421	−0.01952	28,566,900	0.99988
*C_s_,* _0.35,0.1_	0.02082	−0.01621	9,915,150	0.99258
*C_s_,* _0.35,0.2_	0.03044	−0.02369	19,083,800	0.99939
*C_s_,* _0.40,0_	0.01867	−0.0141	11,025,700	0.99498
*C_s_,* _0.40,0.1_	0.02159	−0.017	6,953,770	0.97481
*C_s_,* _0.40,0.2_	0.02472	−0.01719	9,255,250	0.9905

**Table 8 materials-16-00869-t008:** Calculation results for *C_s_*.

Water to Cement Ratio	Load	*Cs* (%)				
		30 d	60 d	90 d	120 d	180 d
0.30	0	0.6033	0.7594	0.8062	0.8707	0.9872
	0.1	0.8618	0.9734	1.11	1.203	1.376
	0.2	0.8916	1.118	1.326	1.568	1.778
0.35	0	0.6392	0.7975	0.9185	1.077	1.286
	0.1	0.8296	1.1273	1.346	1.5	1.748
	0.2	1.021	1.141	1.478	1.734	1.97
0.40	0	0.7717	0.9503	1.146	1.377	1.502
	0.1	1.005	1.329	1.543	1.881	1.941
	0.2	1.147	1.553	1.714	1.868	2.172

**Table 9 materials-16-00869-t009:** Calculation results for *D*.

Water to Cement Ratio	Load	*D* (10^−11^m^2^s^−1^)				
		30 d	60 d	90 d	120 d	180 d
0.30	0	39.601	0.30	0	39.601	0.30
	0.1	43.494	34.531	0.1	43.494	22.570
	0.2	51.889	39.307	0.2	51.889	23.432
0.35	0	29.234	0.35	0	29.234	0.35
	0.1	37.501	29.988	0.1	37.501	18.740
	0.2	50.059	41.301	0.2	50.059	24.023
0.40	0	25.813	0.40	0	25.813	0.40
	0.1	35.395	28.970	0.1	35.395	19.353
	0.2	45.247	39.177	0.2	45.247	27.893

**Table 10 materials-16-00869-t010:** Results and goodness of curve fitting (*D*).

Number	A	m	Relevance
*D* _0.30_ _,0_	9.368 × 10^−8^	−0.36324	0.99836
*D* _0.30_ _,0.1_	1.24 × 10^−7^	−0.39218	0.99837
*D* _0.30_ _,0.2_	6.68 × 10^−8^	−0.35438	0.99836
*D* _0.35,0_	4.00 × 10^−8^	−0.33245	0.99836
*D* _0.35_ _,0.1_	1.24 × 10^−7^	−0.39218	0.99837
*D* _0.35,0.2_	1.50 × 10^−7^	−0.3845	0.99837
*D* _0.40,0_	6.03 × 10^−6^	−0.68085	0.99859
*D* _0.40,0.1_	6.68 × 10^−8^	−0.35438	0.99836
*D* _0.40,0.2_	2.28 × 10^−8^	−0.26446	0.99836

**Table 11 materials-16-00869-t011:** Model parameter settings.

Parameters	Symbols	Numerical Value
Geometric model length	*L_a_*	100 mm
Geometric model width	*L_b_*	150 mm
Diffusion coefficient (m^2^/s)	*D*	Equations (19) to (27)
Initial chloride ion concentration (%)	*C_0_*	0.3216%, 0.4204%, 0.5579%

## Data Availability

Not applicable.

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
