# Peer review of "Study of Chloride Ion Diffusion in Coral Aggregate Seawater Concrete with Different Water–Cement Ratios under Load"

_materials, 2023, doi:10.3390/ma16020869_

Round 1

Reviewer 1 Report

If we want to make conclusions about ions, we should know the exact chemical composition of the cement, its exact fineness, the reason why the amount of cement is exactly the same in relation to the filler, the granular composition of the aggregate, the chemical characterization of the sand and aggregate used.

SEM images, chemical analyzes of materials after tests, X-ray analyzes are missing.

Author Response

请参阅附件。

Reviewer 2 Report

The manuscript's theme is quite interesting, especially regarding the influence of loading on the diffusion of chlorides in concrete structures. However, the work needs to improve the details of the methods used and the discussion of the results. The discussion presented is exhausting and does not explain the observed behaviors. Some specific comments are described below:

1) “Chloride ion erosion” Erosion? Is this term the most correct? I believe that erosion is due to contact with sea water and not exclusively attributed to chlorides.

2) In the introduction it is not clear what the environmental issue is and why it is important to use coral aggregates for the production of concrete. Please enter this information.

3) In addition, it is important to mention in the introduction the impact of these aggregates on the properties of concrete to facilitate the understanding and discussion of the results of chloride diffusion, the focus of the manuscript.

4) In Table 3 it is unclear whether or not the w/c ratio shown considers the amount of water in the pre-saturation stage of the aggregates. Furthermore, the authors must mention the impact of this pre-saturation procedure after molding the specimens. For example, what is the effect of this water on the cement hydration process?

5) “As stated in 93 Table 2, the mixing water was artificial seawater prepared according to ASTM D1141-2013 94 [28].” The artificial seawater was also used for the production of the concretes? From a practical point of view does this make sense? We usually take care not to use contaminated water to produce concrete. Furthermore, what is the impact of the components present in this water on the hydration process and on the properties of concrete? For example, how do chlorides influence hydration kinetics and mechanical properties? These aspects need to be mentioned in the text.

6) “Specimen sizes of 300×100×100mm and 100×100×100mm are used. 15 numbers of 300× 104 100×100mm specimens with the same water-cement ratio were tested for 30, 60, 90, 120, 105 150 and 180 days respectively by artificial seawater erosion. 5 numbers are loaded with 0, 106 5 numbers are loaded with 0.1, 5 numbers are loaded with 0.2. Specimens of 300×100×100mm are 45 in total. 3 numbers of specimens are 100×100×100mm”. Rewrite sentence to improve understanding of information.

7) “A hole is drilled to obtain chloride ion powder at different depths of specimen, and the chloride ion concentration is determined using a rapid chloride ion test.” Enter equipment specifications and test conditions, such as sample preparation.

8) Figure 8 is not mentioned in the text. Moreover, was this test device based on any previous study? Please enter more information about it.

9) Improve the quality and font size of Figure 5.

10) The procedure described in item 2.3.2.1. The sampling method is based on what? This information is essential to the article.

11) “m is the attenuation coefficient of chloride ion diffusion coefficient.” How can this coefficient be determined? And how was it defined in the case of the manuscript?

12) “The free chloride ions content on the concrete surface will gradually grow as the ero-187 sion time increases. After obtaining the chloride ion content at different times, the data 188 can be fitted and expressed as the following power function…” Reference equation.

13) Increase the font size of all graphics in the manuscript.’

14) “The Cf of CASC is determined for 30, 60, 90, 120, 150 and 180 days of the test with  different test environments, and the specific data are shown in the Figure” Which figure?

15) The discussion of the results needs to be revised entirely. Basically, there is a description of the graphs' results and values, without explaining general trends and explanations for the observed behaviors. Furthermore, the results are not compared with the literature. Reading is very difficult and tiring.

16) Present the diffusion coefficient data (Table 8) in a graph for easy visualization.

17) “MATLAB software is used to automatically build the internal coarse and fine aggregate model of CASC based on the characteristics of CASC having a higher porosity than conventional concrete. The model is imported into COMSOL software for simulation calculation. The specific model is shown in Figure 12.” This information does not belong in the results section.

18) “The established CASC model is simulated numerically using COMSOL software, 405 combined with a series of conditions such as initial conditions, boundary conditions and 406 chloride ion diffusion coefficients” Describe in detail all information and boundary conditions adopted in the simulation. This information is essential.

Round 2

Reviewer 2 Report

Accept in present form